# Factors Influencing the Intention to Sign Up for Advanced Care Planning in Day Care for Psychiatric Patients

Yi-Chien Chen [1], Chin-Yu Huang [1] and Chao-Hsien Lee [2,*]

1. Department of Nursing, Taipei Veterans General Hospital, Yuli Branch, Hualien 98142, Taiwan; dp@vhyl.gov.tw (Y.-C.C.); j5053@mail.vhyl.gov.tw (C.-Y.H.)
2. Department of Nursing, Meiho University, Neipu, Pingtung 91202, Taiwan
* Correspondence: x00002167@meiho.edu.tw; Tel.: +886-8-7747168

**Abstract:** (1) Background: Currently, Taiwan has adopted a "person-centered" approach to Advance Care, including Hospice Palliative Care and Advance Decisions, both of which are intended to enhance the right of individuals to choose their own end-of-life care; however, it is extremely challenging and difficult to implement the principle of autonomy for psychiatric patients. (2) Methodology: The aim in this study is to investigate the factors affecting the intention of day ward patients to sign up for hospice and palliative care by using the questionnaire content of the "Survey on Knowledge, Attitude Toward, Experience, and Behavior Intention to Sign Up for Hospice and Palliative Care". A cross-sectional design compliant with STROBE (Strengthening the Reporting of Observational Studies in Epidemiology) was employed. An independent sample *t*-test, Pearson's correlation analysis, and stepwise regression analysis were used to determine the factors influencing the intention of psychiatric patients to sign up for advanced care planning. (3) Results: The relationships between knowledge of and attitude toward advanced care planning, knowledge of and behavior intention to sign up for advanced care planning, and attitude toward and behavior intention to sign up for advanced care planning were all positive (*p*-value < 0.001). The final three most relevant indicators were attitude toward hospice and palliative care, hospitalization of family members during the previous five years, and death of a close friend within the previous five years. (4) Conclusions: The results of the study show that the hospice and palliative care attitude and past experience of psychiatric patients affect the intention to sign up, reminding us that psychiatric patients are at an increased risk of decision-making disability as their illness progresses and that, in addition to initiating the discussion of Advance Care Planning as soon as possible, it may be an opportune time for medical professionals to actively promote Advance Care Planning among their patients.

**Keywords:** mental illness; mental rehabilitation institutions; patient right to autonomy act; advance care planning; advance decisions

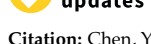



## 1. Introduction

Since the Hospice Palliative Care Act [1] was published on 7 June 2000, hospice and palliative care law has been in effect in Taiwan, enabling an individual to fill out a Hospice Palliative Care form to make a decision on his or her own terminal care. [2]. For this reason, after 16 years of efforts and amendments, Taiwan finally implemented the Patient's Autonomy Act (PAA) on 6 January 2016, in which the wishes of the family members are recognized through the discussion process of Advance Care Planning (ACP). When the disease faced is classified under "end-of-life", "irreversible coma", "permanent vegetative state", "severe dementia", or "other diseases declared by the central authority", the patient can choose the medical care method of "life-sustaining treatment" or "artificial nutrition and fluid feeding" and decide on his or her own medical care wishes from four options, namely "accept", "refuse", "try treatment for a period of time", or "let the medical appointed agent decide", thus completing an Advance Decision (AD) [3]. Respecting the autonomy of a

consenting individual is a fundamental moral principle [4]. The difference between Hospice Palliative Care and Advance Decisions is that a competent individual over the age of 20 can complete a Hospice Palliative Care form with two witnesses (who may not be immediate family members) and submit it to the Ministry of Health and Welfare for registration without cost. An Advance Decision, on the other hand, requires the participation of a medical consultation team and two second-degree relatives or medical appointment agents, and the medical institution may charge a consultation fee [1], regardless of the type of letter of intent, with the intention of enhancing the national's right to choose their own end-of-life care; however, the implementation of the principle of autonomy is extremely challenging and difficult to apply to psychiatric patients [5].

The Ministry of Health and Welfare of Taiwan reported that the number of chronic psychiatric patients in Taiwan increased from 113,992 in 2011 to 133,056 in 2021 [6]; the majority of chronic psychiatric patients are diagnosed with psychosis and have a life expectancy that is 14.5 years lower than the general population [7]. However, they are more likely to be excluded from Advance Care Planning owing to mental disability or loss of mental capability, and they have less autonomy in decision making due to inadequate knowledge [8,9]. The target population for psychiatric day ward services are patients who have been diagnosed by specialists as having psychiatric disorders and rehabilitation potential [10], primarily patients with stable psychiatric symptoms, no risk of self-injury or injury, and rehabilitation capability, despite the fact that these patients have irreversible chronic diseases, but are also unstable due to the long duration of the disease [11]. Therefore, in practice, their condition is frequently labeled as "mental capacity impairment or loss of mental capacity" [12]. Article 12 of the Taiwanese Civil Code specifies that an adult must be at least 20 years old, although individuals who are incapable of expressing or recognizing a decision's meaning owing to mental disability or mental deficiency may seek for a guardianship declaration [13]. In Taiwan, in addition to age and the Mini-Mental State Examination (MMSE) to assess mental and cognitive functions, the declaration of guardianship is the major criterion for defining "capacity" in clinical practice. Unfortunately, patients with psychiatric disorders who may benefit from hospice and palliative care or Advance Care Planning are often automatically excluded from completing a will in the early stages of illness due to cognitive impairment, emotional disorders, communication disorders, or guardianship issues [8,9].

In Taiwan, where Advance Care has been promoted to date, research related to Advance Care Planning for patients with dementia and mental disorders has only been developed gradually since 2019 [14,15]. Due to the late development of research in Taiwan, the evidence base of research on Advance Care Planning for psychiatric patients in Taiwan is currently limited, and their medical care choices and needs are still unclear. To overcome this difficulty, the first step is to determine whether psychiatric patients intend to enroll in hospice and palliative care in order to further determine the acceptability of the Advance Care Planning process and explore the viability of Advance Decision for patients with mental illness. The results of this study will undoubtedly constitute an important advancement and breakthrough for the future development of the Advance Care Planning model for psychiatric patients in Taiwan, which was the impetus for the authors' study. In the field of psychiatric treatment, the day ward is an essential rehabilitation facility for chronic and stable psychiatric patients, as well as a point of reintegration into the community.

Consequently, a day ward in a psychiatric teaching hospital in eastern Taiwan serves as the focus of this study, the objective of which is to compare the knowledge, attitude, and experiences of day ward patients based on their basic demographic characteristics regarding hospice and palliative care, as well as to investigate the factors that influence the intention of day ward patients to sign up for hospice and palliative care.

## 2. Materials and Methods

### 2.1. Study Area and Study Design

Adopting the behavioral theory of the Ajzen Project, with reference to the questionnaire content of the "Survey on Knowledge, Attitude Toward, Experience, and Behavior Intention to Sign Up for Hospice and Palliative Care" conducted by Lin et al. (2011) [16] and Chen (2017) [17], a literature review, the content of the Taiwan Hospice Palliative Care Act, expert recommendations, and authors' experimental design shown in Supplementary File S1, the structural questionnaire of this study was developed, for which the total CVI was 0.975 and the Cronbach's $\alpha$ was 0.762. This study was conducted in a psychiatric hospital in the eastern part of Taiwan, with a total of 130 beds. Psychiatric patient diagnoses included schizophrenia (ICD-10-CM Diagnosis Code F20) and bipolar disorder (ICD-10-CM Diagnosis Code F31). The aim was to investigate the factors affecting the intention of day ward patients to sign up for hospice and palliative care, which had been reviewed by the Clinical Research Ethics Committee of the Antai Medical Care Cooperation Antai-Tian-Sheng Memorial Hospital prior to the study being conducted (trial number: 21-105-B).

### 2.2. Inclusion and Exclusion Criteria

Informed consent and questionnaires were collected by one nurse who was certified in Advanced Care Planning and had completed IRB education, without infringing on the rights and interests of participants and non-participants in day care. Participants were (1) patients aged 20 years or older who received care at the study site day ward from 1 October 2021 to 31 December 2021; (2) those who were conscious and could answer in Taiwanese; (3) those who did not mind the content concerning hospice and palliative care in the questionnaire and did not feel uncomfortable with or afraid of certain words; and (4) considering the reliability and stability of the data obtained, those with cognitive function with a score of $\geq 23$ on the Mini-Mental State Examination based on their medical records. The study excluded (1) those who could not communicate in Mandarin or Taiwanese and those who were incapacitated or under guardianship; (2) those who did not wish to sign the participant consent form; (3) those who were transferred to an acute-care unit during the study period and were hospitalized or settled (including those who were discharged, took more than 7 days of leave in the month and did not show up for day ward, or were automatically discharged); (4) participants who withdrew at any time according to their own wishes; (5) considering the reliability and stability of the data obtained, those with cognitive function with a score of $<23$ on the Mini-Mental State Examination based on their medical records; and (6) those who had a declaration of guardianship of psychiatric patients.

In order to achieve the necessary statistical power and take into account sample attrition, we conservatively estimated that the effect size = 0.8 (big effect size), $\alpha = 0.05$, and power = 0.80. Advanced care planning among mental patients is a novel and culturally sensitive problem in Asia. Although the required sample size was computed to be 52 individuals, we ultimately chose to recruit 73 participants. Sample size estimation was conducted by G*Power (Version 3.1).

### 2.3. Research Hypothesis

**H1.** *Differences in knowledge, attitude, and behavior intention to sign up related to basic individual demographic factors in hospice and palliative care.*

**H2.** *Differences in hospice and palliative care knowledge, attitude, and behavior intention to sign up between care experiences.*

**H3.** *Correlations between age, hospice and palliative care knowledge, attitude, and behavior intention to sign up.*

**H4.** *Factors affecting day ward patients' intention to sign up for hospice and palliative care.*

### 2.4. Questionnaire Design

To determine the present situation of day ward patients in regards to Advance Care Planning, the questionnaire comprised five major sections: basic demographic characteristics, knowledge, attitude, care experience, and behavior intention to participate (Figure 1).

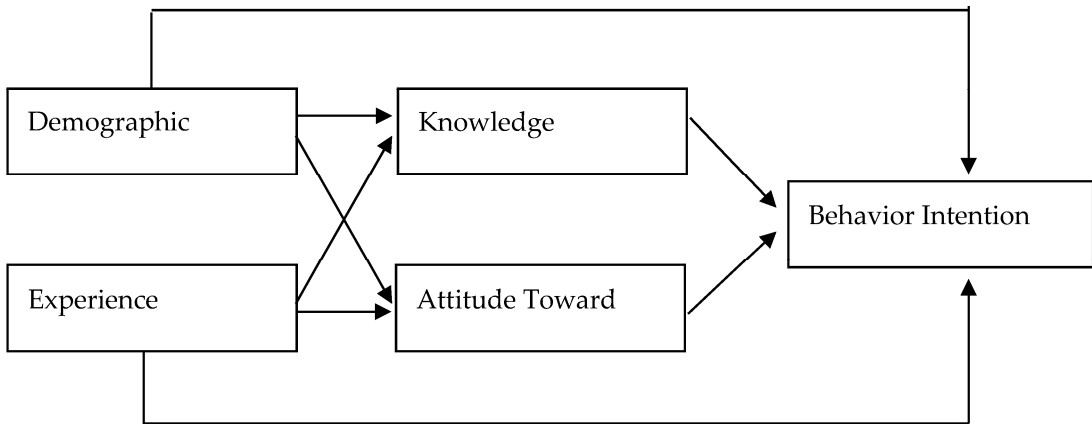

**Figure 1.** Research Framework.

### 2.4.1. Demographic Characteristics

Age, gender, education level, religion, workplace employment, physical co-morbidity, and caregiver experience were included as the 7 categories. The above variables were treated as categorical variables and dummy variables.

### 2.4.2. Knowledge

The Hospice and Palliative Care Knowledge Scale focuses on hospice and palliative care awareness and the legal notion of hospice and palliative care in Taiwan. A total of 13 questions were scored as follows: 1 point for "yes", 0 points for "no", and 0 points for "unclear".

### 2.4.3. Attitude

To understand day ward patients' subjective and objective judgments of their attitude toward hospice and palliative care, in terms of the degree of agreement or disagreement, different scores were assigned to each level according to the inclination of the statements, which were "strongly disagree", "disagree", "no opinion", "agree", and "strongly agree", with a total of 11 questions. The higher the score, the more positive the attitude (e.g., "strongly agree" scores 5, and "agree" scores 4), with a total score range of 11~55.

### 2.4.4. Experience

A total of 9 individual history items are included: whether there was any hospitalization experience in the past 5 years, whether there was any hospitalization of family members in the past 5 years, whether there is any family member who had passed away in the past 5 years, whether there is any close friend who had passed away in the past 5 years, whether there is any discussion about hospice and palliative care, whether there is any discussion about hospice and palliative care with medical personnel, whether there is any participation in hospice and palliative care promotion activities, the patient has heard anything about hospice and palliative care, and whether there is any wish to register for hospice and palliative care with their health insurance card. The total score ranges from 0 to 9 points, with a score of 1 point for "yes" and 0 points for "no".

2.4.5. Behavior Intention

The contents include "Pre-established Hospice and Palliative Care", "No Cardiopulmonary Resuscitation", "No Life Support Medical Treatment", "Appointment of Advance Care Agent", and "Intent to Register Hospice and Palliative Care on Health Insurance Card". The scores are calculated by scoring 1 point for "Very Unlikely", 2 points for "Unlikely", 3 points for "Uncertain", 4 points for "Likely", and 5 points for "Very Likely", with a total score range of 5~25 points.

*2.5. Data Analysis*

Advance Care consultation process must be performed by trained personnel. Therefore, the data were gathered by nursing personnel with expertise in Advance Care Planning, and we provided and obtained the written informed consent to participate from all the participants in this study, analyzed using IBM SPSS Statistics, Armonk, NY, USA (Version 22). Descriptive statistics (subdivision, percentage, mean, and standard deviation) were used to present the patient characteristics of the day ward; inferential statistics (independent sample *t*-test, Pearson correlation analysis, and stepwise regression analysis) were utilized to predict and comprehend the factors influencing the intention of day ward patients to enroll in hospice and palliative care.

**3. Results**

*3.1. Analysis of Knowledge, Attitude, and Behavior Intention in Relation to Demographic Factors in Hospice and Palliative Care*

Out of 75 questionnaires returned, 2 of them were discarded because they were completed halfway. Thus, the total number of psychiatric patients in the day ward was 73. The mean ± SD age of the participants was 57.48 ± 11.33 years (minimum age was 21 years, maximum age was 79 years), and participant diagnoses included schizophrenia (62 participants, or 84.9%) and bipolar disorder (11 participants, or 15.1%). The analysis revealed that the total scores of hospice and palliative care knowledge, attitude, and behavior intention to sign up for those with caregiving experience were higher than the scores for those with no caregiving experience. The mean score for the total attitude toward hospice and palliative care for those with religious beliefs was also higher than that of those without religious beliefs, and all were statistically significant. There were no statistically significant differences in the analysis of patients' knowledge of, attitude toward, and behavior intention to sign up for hospice and palliative care with respect to other demographic attributes (Table 1).

*3.2. Analysis of Care Experience in Relation to Hospice and Palliative Care Knowledge, Attitude, and Behavior Intention to Sign Up*

In terms of attitude toward hospice and palliative care, the mean scores of those who answered "whether they have been hospitalized in the past five years" and "have talked to someone about hospice and palliative care" affirmatively were both higher than those without experience, with statistically significant differences. Regarding the intention to sign up for hospice and palliative care, the mean scores of those who answered "whether a family member had been hospitalized in the past five years", "whether a close friend had passed away in the past five years", "have talked to someone about hospice and palliative care", "have heard about hospice and palliative care", and "have hospice and palliative care registered on their health insurance card" affirmatively were all significantly higher than those with no experience, as demonstrated in Table 2.

**Table 1.** Analysis of Demographic Factors in Hospice and Palliative Care Knowledge, Attitude, and Behavior Intention to Sign Up.

| Factors | N | Knowledge | | | Attitude Toward | | | Behavior Intention | | |
|---|---|---|---|---|---|---|---|---|---|---|
| | | Mean (SD) | *t* | *p*-Value | Mean (SD) | *t* | *p*-Value | Mean (SD) | *t* | *p*-Value |
| **Age** | | | | | | | | | | |
| >60 years old | 38 | 6.87 (3.22) | 0.20 | 0.841 | 36.95 (7.03) | 0.32 | 0.752 | 15.79 (5.06) | 0.18 | 0.859 |
| ≥60 years old | 35 | 6.71 (3.32) | | | 36.46 (6.07) | | | 15.60 (3.90) | | |
| **Gender** | | | | | | | | | | |
| Male | 49 | 6.47 (3.25) | −1.23 | 0.224 | 36.35 (6.48) | −0.68 | 0.499 | 15.94 (4.56) | 0.65 | 0.520 |
| Female | 24 | 7.46 (3.20) | | | 37.46 (6.74) | | | 15.21 (4.47) | | |
| **Education** | | | | | | | | | | |
| Below junior high school | 22 | 6.36 (2.36) | −0.74 | 0.461 | 36.32 (3.88) | −0.42 | 0.673 | 16.14 (3.82) | 0.54 | 0.590 |
| High school (vocational) or above | 51 | 6.98 (3.57) | | | 36.88 (7.43) | | | 15.51 (4.80) | | |
| **Religion** | | | | | | | | | | |
| No | 10 | 5.90 (3.63) | −0.94 | 0.352 | 32.40 (5.95) | −2.31 * | 0.024 | 13.40 (5.19) | −1.76 | 0.083 |
| Yes | 63 | 6.94 (3.19) | | | 37.40 (6.41) | | | 16.06 (4.33) | | |
| Workplace Employment | | | | | | | | | | |
| No | 37 | 6.49 (3.36) | −0.82 | 0.415 | 35.95 (7.01) | −1.02 | 0.314 | 15.57 (4.89) | −0.25 | 0.803 |
| Yes | 36 | 7.11 (3.14) | | | 37.50 (6.02) | | | 15.83 (4.15) | | |
| **Physical Co-morbidity** | | | | | | | | | | |
| No | 25 | 6.44 (3.57) | −0.67 | 0.505 | 35.60 (6.61) | −1.05 | 0.298 | 15.76 (4.61) | 0.08 | 0.934 |
| Yes | 48 | 6.98 (3.09) | | | 37.29 (6.50) | | | 15.67 (4.51) | | |
| **Caregiver Experience** | | | | | | | | | | |
| No | 40 | 6.03 (3.49) | −2.29 * | 0.025 | 34.93 (6.68) | −2.68 ** | 0.009 | 14.33 (4.99) | −3.02 ** | 0.003 |
| Yes | 33 | 7.73 (2.70) | | | 38.88 (5.75) | | | 17.36 (3.19) | | |

Note: * *p*-value < 0.05, ** *p*-value < 0.01.

**Table 2.** Analysis of care experience in hospice and palliative care knowledge, attitude, and behavior intention to sign up.

| Experience | | N | Knowledge | | | Attitude Toward | | | Behavior Intention | | |
|---|---|---|---|---|---|---|---|---|---|---|---|
| | | | Mean (SD) | t | *p*-Value | Mean (SD) | t | *p*-Value | Mean (SD) | t | *p*-Value |
| Hospitalization experience within the past five years | No | 26 | 6.04 (3.54) | −1.49 | 0.140 | 34.04 (7.86) | −2.42 * | 0.021 | 14.69 (5.61) | −1.27 | 0.210 |
| | Yes | 47 | 7.21 (3.04) | | | 38.19 (5.21) | | | 16.26 (3.73) | | |
| Hospitalization events of family members in the past five years | No | 44 | 5.95 (3.20) | −2.85 ** | 0.006 | 36.00 (6.51) | −1.15 | 0.255 | 14.61 (4.90) | −2.85 ** | 0.006 |
| | Yes | 29 | 8.07 (2.94) | | | 37.79 (6.55) | | | 17.34 (3.28) | | |
| Any death of a family member in the past five years | No | 40 | 6.00 (3.60) | −2.46 * | 0.017 | 36.28 (6.73) | −0.63 | 0.533 | 15.13 (5.20) | −1.2 | 0.234 |
| | Yes | 33 | 7.76 (2.49) | | | 37.24 (6.37) | | | 16.39 (3.45) | | |
| Any death of a close friend in the past five years | No | 50 | 6.30 (3.44) | −1.96 | 0.055 | 36.44 (3.89) | −0.52 | 0.603 | 14.78 (4.74) | −2.67 ** | 0.009 |
| | Yes | 23 | 7.87 (2.55) | | | 37.30 (5.82) | | | 17.70 (3.24) | | |
| Engaged in discussions regarding hospice and palliative care with others | No | 52 | 6.44 (3.50) | −1.73 | 0.090 | 35.58 (5.83) | −2.41 * | 0.019 | 14.73 (4.52) | −3.05 ** | 0.003 |
| | Yes | 21 | 7.67 (2.37) | | | 39.52 (7.46) | | | 18.10 (3.55) | | |
| Engaged in discussions regarding hospice and palliative care with medical professionals | No | 56 | 6.63 (3.50) | −1.02 | 0.312 | 35.91 (6.61) | −1.94 | 0.057 | 15.21 (4.81) | −1.69 | 0.096 |
| | Yes | 17 | 7.35 (2.21) | | | 39.35 (5.73) | | | 17.29 (2.93) | | |
| Participated in "Hospice and Palliative Care" promotional activities | No | 56 | 6.55 (3.48) | −1.45 | 0.155 | 37.00 (6.71) | 0.68 | 0.499 | 15.27 (4.63) | −1.49 | 0.140 |
| | Yes | 17 | 7.59 (2.24) | | | 353.76 (6.03) | | | 17.12 (3.90) | | |
| Had heard about "hospice and palliative care" | No | 32 | 5.50 (3.87) | −3.00 ** | 0.004 | 35.47 (6.03) | −1.45 | 0.153 | 14.25 (4.91) | −2.51 * | 0.014 |
| | Yes | 41 | 7.80 (2.24) | | | 37.68 (6.83) | | | 16.83 (3.87) | | |
| Registered wishes for hospice and palliative care on health insurance card | No | 59 | 6.63 (3.51) | −1.36 | 0.182 | 36.44 (6.62) | −0.73 | 0.470 | 15.19 (4.73) | −2.03 * | 0.046 |
| | Yes | 14 | 7.50 (1.70) | | | 37.86 (6.32) | | | 17.86 (2.63) | | |

Note: * *p*-value < 0.05, ** *p*-value < 0.01.

### 3.3. Correlation of Age with Hospice and Palliative Care Knowledge, Attitude, and Behavior Intention to Sign Up

Regarding the correlation of age with hospice and palliative care knowledge, attitude, and behavior intention to sign up, no correlation existed of age with hospice and palliative care knowledge, attitude, and behavior intention to sign up. However, the correlation coefficients were 0.491 ($p$-value < 0.001) between hospice and palliative care knowledge and attitude, 0.416 ($p$-value < 0.001) between hospice and palliative care knowledge and behavior intention to sign up, and 0.615 ($p$-value < 0.001) between hospice and palliative care attitude and behavior intention to sign up, which showed positive and statistically significant correlations.

### 3.4. Factors Affecting Day Ward Patients' Hospice and Palliative Care Behavior Intention to Sign Up

For the purpose of constructing a regression analysis model of day ward patients' intention to sign up for hospice and palliative care, statistically significant factors related to behavior intention to sign up for hospice and palliative care were considered, which included the following: experience as a caregiver, whether a family member had been hospitalized in the past five years, whether a close friend had passed away in the past five years, having talked to someone about hospice and palliative care, having heard about hospice and palliative care, the desire to register hospice and palliative care on a health insurance card, knowledge of hospice and palliative care, and attitude toward it, among other factors. A stepwise regression analysis was performed by model selection methods. Ultimately, three important factors were identified as statistically significant and were retained in the model: hospice and palliative care attitude, whether a family member had been hospitalized in the past five years, and whether a close friend had passed away in the past five years. This indicates that the model has no co-linearity problems, indicating that the model has an acceptable range of applicability (Table 3).

**Table 3.** Factors affecting day ward patients' hospice and palliative care behavior intention to sign up.

| Factors | Unstandardized Coefficients | | | $t$ | $p$-Value | VIF |
|---|---|---|---|---|---|---|
| | ß | SE | 95% CI | | | |
| Intercept | −4.19 | 2.52 | [−9.13, 0.75] | −1.66 | 0.102 | |
| Attitude | 0.40 | 0.06 | [0.28, 0.52] | 6.57 *** | <0.001 | 1.02 |
| **Hospitalization events of family members in the past five years** | | | | | | |
| Yes vs. No | 1.65 | 0.81 | [0.06, 3.24] | 2.02 * | 0.047 | 1.05 |
| **Any death of a family member in the past five years** | | | | | | |
| Yes vs. No | 2.27 | 0.85 | [0.60, 3.94] | 2.67 ** | 0.009 | 1.03 |

Note: $R^2$ = 0.48, ADJ($R^2$) = 0.46, * $p$-value < 0.05, ** $p$-value < 0.01, *** $p$-value < 0.001.

## 4. Discussion

In this study cohort, there was no difference in the knowledge of, attitude toward, and behavior intention to sign up for hospice and palliative care among patients with psychiatric disorders who also had physical co-morbidities, which suggests that participants may have little knowledge of their own health status or may not be able to readily articulate their discomfort and thus neglect their underlying chronic illness. [9,18,19]. In a study on the implementation of Advance Care Planning for psychiatric patients in a hospital in northern Taiwan, it was found that the choice of Advance Care was not influenced by gender [15], and the results of this study also show similarly that gender did not affect the intention to sign up. Furthermore, the attitude of religious individuals towards hospice and palliative care was more positive, consistent with Van Wijman et al.'s study [20]. Among those who had previous experience in caregiving, there were significant differences in hospice

and palliative knowledge, attitude, and behavior intention to sign up, which may lead to the hypothesis that their personal experience in caregiving may drive them to think about future situations in which they are in need of care from others and motivate them to participate in Advance Care Planning, which echoes the findings of Lin et al. [12].

In further exploring the effect of caregiving experience on hospice and palliative care knowledge, attitude, and behavior intention to sign up, "whether a family member has been hospitalized in the past five years" and "heard about hospice and palliative care" showed a positive correlation with hospice and palliative care knowledge and behavior intention to sign up. As hypothesized based on the authors' empirical experience, day ward patients possess rehabilitation potential and themselves have a demand for knowledge when stimulated by external factors. Medical care teams are encouraged to eliminate stereotypes of psychiatric patients' lack of comprehension and provide timely opportunities for gaining hospice and palliative care knowledge [12]. Having "talked to someone about hospice and palliative care" affects both attitudes and intention to sign up for hospice and palliative care, which reaffirms the importance of establishing discussions concerning Advance Care with individuals with mental illness and encouraging them to express their self-perceptions and emotions. The results of this study incidentally demonstrate no difference in knowledge, attitude, and behavior intention to sign up between "having talked about hospice and palliative care with medical staff" and "having participated in hospice and palliative care promotion activities", echoing the results of the authors' study on the impact of intervention models on Advance Care from a caregiver perspective in Eastern Hualien, Taiwan, in 2018 [21]. As a corollary, regardless of the subject's illness or societal position, health care professionals are encouraged to communicate Advance Care in terms that are comprehensible to the subject, and to confirm that the subject truly understands the implications of Advance Care [15]. In summary, prior experiences of psychiatric patients are closely related to hospice and palliative care knowledge, attitude, and behavior intention to sign up, whereby health care teams are advised to provide individualized education and advocacy based on past experiences [12]. It is arguably challenging to collect cases from psychiatric patients as a cohort, coupled with the lack of supporting evidence in the current study on the impact of Advance Care on psychiatric patients, as well as a paucity of supporting data on care experiences, suggesting that future research efforts are warranted.

Cal et al. (2011) conducted an analysis of the participants in an Iowa hospice, whose average age was 80.4 years. There, age was identified as one of the factors influencing the behavior intention to sign up, and those who were psychiatric patients were more willing to receive aggressive treatment [9], contrary to our hypothesis. There is no correlation of knowledge and attitude with signing up or of hospice and palliative care knowledge and attitude with age, presumably due to the low mean age of the study cohort (mean 57.48 years), consistent with that of Huang et al. in Taiwan [15], with similar mean age of the participants. However, there is an existing positive correlation of knowledge and attitude with behavior intention to sign up, as well as between attitude and behavior intention to sign up.

Towards the conclusion of this study, the most influential factors in the behavior intention to sign up were attitudes and care experiences, in particular, whether or not a family member had been hospitalized in the past five years and whether or not a close friend had passed away in the past five years. Unfortunately, ultimately, the knowledge aspect of hospice and palliative care in the stepwise regression did not have an impact on the behavior intention to sign up, with impaired or diminished mental capacity prone to exclusion from Advance Care Planning. Consequently, a majority lacked knowledge related to hospice and palliative care and Advance Care [8,9]. This study unexpectedly identified that psychiatric patients and dementia patients share the same mental deficits attributable to their diseases, with similar disease trajectories, which exhibit fluctuating and prolonged disease profiles, resulting in an increased risk of decision-making impairment corresponding to disease progression. Beyond the early initiation of Advance Care Planning

discussions, opportunities may arise, when a patient's family member is hospitalized or experiences the passing of a close friend, for medical staff to actively engage in Advance Care discussions and further assist with the signing of a letter of intent.

Currently, there are no consistent care guidelines for individuals with psychiatric disorders in Advance Care Planning, with varying findings, but what is certain is that, compared to the general healthy population, individuals with mental illness are less likely to access and sign Advance Care decisions in writing, be it hospice and palliative care or Advance Care Planning. The central purpose of Advance Care Planning is to promote mental health care in Taiwan, including care for psychiatric patients [22,23].

## 5. Conclusions

Overall, to evaluate the research hypotheses, the results of this study show correlations among psychiatric patients between hospice and palliative care knowledge and attitude, between knowledge and behavior intention, and between attitude and behavior intention, whereas age was not a factor affecting intention to sign up.

Reflecting on the current practice in Taiwan, although the implementation of the 2005 Mental Capacity Act in the United Kingdom adheres to the five principles of not presuming that persons with mental illness are intellectually impaired, most medical professionals remain subconsciously confined by the framework of a psychiatric diagnosis, coupled with the fact that Taiwan's internationally acclaimed universal health insurance does not apply to Advance Care Planning clinics, with fees for consultation varying by hospital. For this reason, at present, our current practice of promoting "person-centered" medical care is in the form of Hospice Palliative Care and Advance Decision letters of intent. For medical staff, individuals who wish to participate in Advance Care Planning, and their families, if they wish to participate in the medical care discussion process, there is ample potential for further efforts. With this said and in mind, this is indeed an inspiration for our future efforts.

## 6. Suggestions and Recommendations

In recent years, Advance Care interventions for patients with dementia have flourished in Taiwan. In contrast, research on Advance Care Planning for psychiatric patients is still in its infancy, requiring more research and clinical practice in the future to identify the implementation constraints and obstacles, as well as to develop appropriate Advance Care guidelines for psychiatric patients. In light of the above factors, future research studies are recommended to utilize mixed methods of research and analysis to better illustrate the results of studies, given the lack of empirical validity of psychiatric patients as a cohort. In addition, a group dynamics model may be employed to compare the intervention models of the Advance Care health care groups to assess whether it is possible to increase the knowledge, attitude, and behavior intention to sign up among patients with psychiatric disorders and to develop approaches to facilitate their understanding of Advance Care descriptions.

This study was STROBE-compliant and followed all reporting guidelines for cross-sectional studies. It is a model for future cross-sectional studies. Supplementary File S1 show the specific recommendations of STROBE, which were met.

## 7. Research Limitations

This study exhibits three limitations. Firstly, the data were obtained only from patients with psychiatric disorders in day wards of a psychiatric hospital in eastern Taiwan, which led to a comparably limited sample size due to participants' disorders, comprehension, and ability to complete questionnaires, thus making it undesirable to extrapolate inferences and interpretations to the entire population of patients with mental illness. Secondly, this study was analyzed through questionnaire data. Thus, the reliability and validity of the results are susceptible to respondents' mental and physical conditions, beliefs, and social contexts at the time of questionnaire completion, and therefore, the results of the study are

limited to those collected by this research instrument. Thirdly, as mentioned in this study, research on psychiatric patients is of limited evidentiary validity due to the difficulties in admission of cases, and thus, the results of this study can only represent correlations, not causal effects.

**Supplementary Materials:** The following supporting information can be downloaded at: https://www.mdpi.com/article/10.3390/nursrep13020076/s1, File S1: Supplementary documents; File S2: Data.

**Author Contributions:** Conceptualization, Y.-C.C. and C.-Y.H.; methodology, Y.-C.C. and C.-H.L.; software, C.-H.L.; validation, Y.-C.C. and C.-Y.H.; formal analysis, C.-H.L.; investigation, Y.-C.C. and C.-Y.H.; resources, Y.-C.C. and C.-Y.H.; data curation, Y.-C.C. and C.-Y.H.; writing—original draft preparation, Y.-C.C. and C.-H.L.; writing—review and editing, Y.-C.C. and C.-H.L.; visualization, Y.-C.C. and C.-H.L.; supervision, Y.-C.C. and C.-H.L.; project administration, Y.-C.C. and C.-Y.H.; funding acquisition, Y.-C.C. and C.-Y.H. All authors have read and agreed to the published version of the manuscript.

**Funding:** This research was funded by National Army Veterans Affairs Council.

**Institutional Review Board Statement:** The study was conducted in accordance with the Declaration of Helsinki and approved by the Institutional Review Board (or Ethics Committee) of the Antai Medical Care Cooperation Antai-Tian-Sheng Memorial Hospital (protocol code, 21-105-B, and date of approval, 5 October 2021).

**Informed Consent Statement:** Informed consent was obtained from all subjects involved in the study.

**Data Availability Statement:** Data are contained within the article or Supplementary Materials.

**Acknowledgments:** We thank the National Army Veterans Affairs Council for their research funding and the director of the Taipei Veterans General Hospital Yuli Branch for their support in the successful realization of this study.

**Conflicts of Interest:** The authors declare no conflict of interest.

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
