# Peer review of "Factors Influencing the Intention to Sign Up for Advanced Care Planning in Day Care for Psychiatric Patients"

_nursrep, doi:10.3390/nursrep13020076_

Round 1

Reviewer 1 Report

In my opinion, the title is misleading. If I well understood, it should be “An Investigation of Correlated Factors for Intention to Sign Up for hospice and palliative care among Psychiatric Patients in A Day Care Center”. Alternatively, “Factors influencing the intention to sign up for an Advanced Care Planning in day care for psychiatric patients”. Get inspired!

While the content is stimulating, the Introduction is too long. For sure, I would avoid the DNR discussion. Then, Figure 1 should be put in supplementary material (or deleted). In addition, the text is focused only on Taiwan. Are there any experiences in other Countries? An international outlook is preferred.

Regarding the methods, please adhere to a checklist from the Enhancing the QUAlity and Transparency Of health Research (EQUATOR). You can also add the checklist as supplementary material when submitting the manuscript.

In addition, Discussion is too long. The readers will tire of it.

Reviewer 2 Report

Dear authors, 

Congratulations on the significant work. Few comments would be:

- Separate the theoretical framework from the design, instrumentation and setting description. Add separate section for each hand elaborate on each one. 

- Specify the exact design of the study in the paper.

- Clarify the language of the questionnaire and cultural validation procedures. 

- add sample size calculation taking into consideration power, effect size and number of hypotheses. 

- Add response rate and management of non-respondents

- Add confidence intervals whenever possible in the results

Reviewer 3 Report

Lines 2-4: The title of the article could be shortened.

Line 88: may need to either change tense to indicate last year, or update information to meet publishing dates.

Lines 157-163: Did you answer the hypotheses? 

Line 211: Please include the version of SPSS: IBM SPSS Statistics (Version 22).

Please clarify what "and that those with severe psychiatric patients" means. (Lines 401-02; 397-98; 370-71; 333-34; 24-25) are just a few places this can be found. It is unclear what you are staying???

Line 390: . . . "patients with psychiatric is still. . ." What are you saying here?

Line 392: Please clarify or complete the thought: . . ."for patients with psychiatric."

  1.  

  1.  

  2.  

Reviewer 4 Report

This is a very important topic 

I am here with some minor recommendations to improve the quality of the article

The introduction is very descriptive. It will be a good idea to be critical and provide some more literature that discusses the variables understudy.  

Researchers need to elaborate more on the ethical aspects. For example,  who did the consent form? is the participants capacitated or incapacitated to provide consent ..etc  

provide sample size calculation 

Reviewer 5 Report

1.    The purpose of this research is to examine the end-of-life attitudes and decisions of psychiatric patients in a day care center in Taiwan. The results of the study show that positive hospice and palliative attitudes and past experiences of psychiatric patients influence the intention to sign up for advance care planning. The manuscript is clear, relevant for the field and presented in a well-structured manner. 

2.      The cited references are mostly recent publications (within the last 5 years) and relevant.

3.      The manuscript scientifically sound and the experimental design appropriate to test the hypothesis.

  1. In the materials, methods and results section the data are incomplete and should be added:

o   Please add the age data of the sample to the table of the demographic factors (Table 1)

o   Please explain the following data: single 54, married 19 (what is the background of these data? the psychiatric disorders?)

    • It is not clear, what psychiatric disorders are present in the sample (in percentage of disorders)

o 

5.      The figures and tables appropriate, they properly show the data, except the Appendix A (Survey Questionnaire)it is difficult to follow the text

6.      Please to explain the term „Supported decision making” in details in the Discussion section (page 10)

7.      The conclusions are consistent with the evidence and arguments presented, but please complete them with the conlusions of the over mentioned data and terms

8.      The ethics statements and data availability statements are adequate.
